# Participant and informant memory-specific cognitive complaints predict future decline and incident dementia: Findings from the Sydney Memory and Ageing Study

**Katya Numbers**[1]*, **John D. Crawford**[1], **Nicole A. Kochan**[1], **Brian Draper**[1,2], **Perminder S. Sachdev**[1,2], **Henry Brodaty**[1,2]

1 CHeBA (Centre for Healthy Brain Ageing), School of Psychiatry, University of New South Wales, Kensington, NSW, Australia, 2 Dementia Collaborative Research Centre, University of New South Wales, Kensington, NSW, Australia

* k.numbers@unsw.edu.au

## Abstract

Subjective Cognitive Complaints (SCCs) may represent one of the earliest stages of preclinical dementia. The objective of the present study was to extend previous work by our group to examine the relationship between participant-reported and informant-reported memory and non-memory SCCs, cognitive decline and incident dementia, over a six-year period. Participants were 873 community dwelling older adults ($M_{age}$ = 78.65, SD = 4.79) without dementia and 843 informants (close friends or family) from the Sydney Memory and Ageing Study. Comprehensive neuropsychological testing and diagnostic assessments were carried out at baseline and biennially for six years. Linear mixed models and Cox proportional hazard models were performed to determine the association of SCCs, rate of cognitive decline and risk of incident dementia, controlling demographics and covariates of mood and personality. Participant and informant *memory*-specific SCCs were associated with rate of global cognitive decline; for individual cognitive domains, participant memory SCCs predicted decline for language, while informant memory SCCs predicted decline for executive function and memory. Odds of incident dementia were associated with baseline participant memory SCCs and informant memory and non-memory SCCs in partially adjusted models. In fully adjusted models, only informant SCCs were associated with increased risk of incident dementia. Self-reported memory-specific cognitive complaints are associated with decline in global cognition over 6-years and may be predictive of incident dementia, particularly if the individual is depressed or anxious and has increased neuroticism or decreased openness. Further, if and where possible, informants should be sought and asked to report on their perceptions of the individual's memory ability and any memory-specific changes that they have noticed as these increase the index of diagnostic suspicion.

**Data Availability Statement:** The terms of consent for research participation stipulate that an individual's data can only be shared outside of the MAS investigators group if the group has reviewed and approved the proposed secondary use of the data. This consent applies regardless of whether data has been de-identified. Access is mediated via a standardised request process managed by the CHeBA Research Bank, who can be contacted at ChebaData@unsw.edu.au, or via the first author's contact details at k.numbers@unsw.edu.au.

**Funding:** HB and PS received the following grants to support this work: three National Health & Medical Research Council (NHMRC) of Australia Program Grants (ID350833, ID568969, APP1093083). https://www.nhmrc.gov.au/funding. The funders had no role in study design, data collection and analysis, decision to publish, or preparation of the manuscript.

**Competing interests:** The authors have declared that no competing interests exist.

## Introduction

Subjective cognitive complaints (SCCs) relate to an individual's self-experience of cognitive decline and are currently a core criterion in the diagnosis of mild cognitive impairment (MCI) [1, 2]. SCCs can be self-reported or reported by those close to an individual and have the potential to capture quickly and easily everyday cognitive and memory problems that may not be detected by standardized neuropsychological tests. Supporting this, SCCs, in the absence of impaired cognitive performance, have been related to the presence of Alzheimer's disease (AD) biomarkers such as amyloid plaques in the brain and tau proteins found in cerebral spinal fluid [3, 4]. For this reason, SCCs are increasingly considered by some researchers to be the earliest detectable, pre-MCI stage of AD [5]. However, despite their potential utility, there have been inconsistent views regarding the reliability of self-reported decline among individuals with cognitive and memory impairment. That is, while some studies have found a relationship between SCCs and cognitive performance [e.g., 6], others studies have not [e.g., 7]. For example, one community-based cohort study reported that only 6% of memory-impaired individuals self-reported cognitive impairment, but that over 90% of those who did complain did not present objective impairments [8]. These conflicting findings have led some to question whether SCCs should be a core diagnostic criterion for MCI and provide little clarity about whether SCCs are useful in detecting prodromal dementia.

Differences in findings across studies may well result from heterogenous methodologies. These variations have been differentially attributed to how SCCs are operationalised and measured [9], how cognitive impairment is operationalised and measured [10], which criteria are used to make an MCI or dementia diagnosis [6], where the sample is recruited from [11], if subjective complaints capture memory specifically or cognition generally [12], and whether an informant is asked to independently corroborate such complaints [13].

Another explanation for competing results across studies may be the common influence of mood and certain personality traits on complaining behaviours in general [14]. Specifically, SCCs are reliably exacerbated in individuals scoring higher on measures of depressive and anxious symptomology [15] as well as those who score higher on neuroticism and lower on openness and conscientiousness personality traits [16]. Increasingly, evidence indicates that anxiety and depressive symptoms may be account for the overreporting of memory and cognitive impairment among individuals with intact cognitive performance [17, 18]. In summary, the reliability of self-reports can be compromised by mood and personality. However, given the potential utility of SCCs for early detection and interventions, further investigation of the relationship between SCCs and cognitive decline is warranted while accounting for potential confounders such as mood and personality.

Previous work by our group has explored the relationships among self- and informant-reported SCCs, mood, personality, and cognitive status in the well-characterised Sydney Memory and Ageing Study, both cross-sectionally [19] and longitudinally [20], though the latter only looked two years from baseline. In the original cross-sectional analyses, neither participant nor informant memory or non-memory SCCs were strongly associated with objective cognitive performance or current clinical diagnoses. Rather, mood and personality were much more strongly associated with cognitive performance and clinical status [19]. Two-years later, after controlling for mood and personality, participant-reported SCCs were not associated with cognitive decline across domains or functional decline for activities of daily living. However, both participant and informant *memory*-specific baseline complaints did show a weak association with incident dementia after two years [20]. Importantly, over longer periods participant-reported SCCs in community-based cohorts have been found to be better predictors of cognitive changes and diagnostic progression to dementia [21].

Given ongoing questions relating to the relationship of SCCs with cognitive decline and incident dementia, and questions about whether SCCs' relationship to cognitive decline and risk of dementia strengthens over time, the present paper extends our previous work to consider the associations between SCCs, cognitive decline, and incident dementia over six-years. More specifically, we explore both participant and informant memory and non-memory SCCs and their ability to predict decline in memory and non-memory domains, as well as their ability to predict incident dementia while controlling for mood and personality. Further, because we now consider a longer follow-up, we use linear mixed models and Cox regressions analyses, which are more robust to non-random attrition bias and allow us to consider the whole sample where previous work by our group considered completers only at two-year follow-up.

## Materials and methods

### Participants

Community-dwelling older adults aged 70–90 years, living in the eastern suburbs of Sydney, Australia, were selected via the electoral roll and invited to participate in the Sydney Memory and Ageing Study (MAS). Of 8,914 individuals invited to participate, 1,037 participants were included in the baseline sample. Inclusion criteria were the ability to speak and write English sufficiently well to complete a psychometric assessment and self-report questionnaires. Exclusion criteria included any major psychiatric diagnoses, acute psychotic symptoms, or a current diagnosis of multiple sclerosis, motor neuron disease, developmental disability, progressive malignancy, or dementia. Additional exclusion criteria included a Mini-Mental State Exam (MMSE) [22] score of < 24, adjusted for age, education and a non-English speaking background at baseline. More detailed methods of recruitment and baseline demographics have been previously published [23]. For the current study, participants were also excluded if they were not able to speak English at a basic conversational level by the age of 9 (N = 164) because of the questionable validity of using normative data based on persons of English-speaking background to determine impairment in these individuals [24]. Of the 873 participants included in the present study, 841 (96.3%) had an informant. Informants were nominated by the participants and answered questions relating to the participant's memory, thinking, and daily functioning. Informants were required to have at least 1 hour of contact with the participant per week; on average they had 8.3 hours of weekly contact. All participants and informants provided written consent to participate in this study, which was approved by the University of New South Wales Human Ethics Review Committee (HC 05037, 09382, 14327).

### Subjective cognitive complaints

Participants completed 24 SCC questions at baseline, which covered both memory (15 questions) and nonmemory (9 questions) cognitive domains. Of the 24 SCC questions, 18 were developed locally by the study group [19] and the remaining 6 were taken from the Memory Complaint Questionnaire (MAC-Q) [25], which asked participants to rate themselves compared with 5-years ago on several everyday memory tasks. At baseline, informants completed 19 SCC questions about the participants, comprised of 15 memory and 4 nonmemory questions. Of the 19 informant SCC questions, 3 were locally developed by the study group [19]; 13 were from a modified version of the short Informant Questionnaire on Cognitive Decline in the Elderly (IQCODE) [26], which asked informants to rate participant's current ability on memory and nonmemory domains compared with 5-years ago (modified from the standard "10 years" ago), and 3 questions targeting functional ability were from the General Practitioner Assessment of Cognition (GPCOG) [27]. From these 43 SCC questions, 4 composite indices were created based on the source (participant or informant) and the nature (memory or

nonmemory) of the complaint. Each SCC item was scored 1 or 0, depending on whether the complaint was endorsed or not and scores were summed for each index to create: participant memory SCCs, participant nonmemory SCCs, informant memory SCCs, and informant nonmemory SCCs. Individual questions within each index can be found in the S1 Appendix.

## Objective cognitive performance

Cognitive performance was assessed using a comprehensive neuropsychological test battery comprised of 10 tests that measured the domains of attention, language, executive function, visuospatial ability, memory and verbal memory. Domain and global cognition composites are presented as standardized z-scores as follows. Raw test scores were first converted to z-scores using the means and standard deviations (SDs) of a reference group comprised of 723 MAS participants classified as cognitively healthy at baseline (native English speakers with a Mini-Mental State Examination score of 24 or above, no evidence of dementia or current depression, no history of delusions or hallucinations, and no major neurological disease, or significant head injuries). Composite domain scores were formed by averaging the z-scores of the component tests. Global cognition scores at baseline and at 6 years were calculated by averaging the domain scores. All domain and Global Cognition scores were standardized against the mean and SD (0 and 1 respectively) of the baseline reference group. More details about how cognitive domain and global cognition scores were calculated, and which tests comprised each cognitive domain, can be found in the S3 Appendix and S1 Table.

## Demographics, affective and personality measures

Demographic information, including age, sex, education, and native-English speaking status, was collected at baseline as were participants' scores on the Goldberg Anxiety Scale (GAS) [28], the 15-item version of the Geriatric Depression Scale (GDS) [29], and the Neuroticism, Openness, and Conscientiousness scales of the NEO-Five Factor Inventory (NEO-FFI) [30].

## Clinical diagnoses

At baseline and at each two-year follow-up, individuals who met the following criteria were brought to a consensus review meeting during which at least three clinicians from a panel of neuropsychiatrists, psychogeriatricians, and neuropsychologists discussed all available clinical, neuropsychological, laboratory and imaging data to make a diagnosis for each individual. Participants were brought to consensus review meetings if they scored at least 1.5 SDs below published normative data on two cognitive measures or showed reduced neuropsychological scores on one cognitive measure and elevated (i.e. poorer) scores on informant-reported activities of daily living.

As dementia was an exclusion criterion for entry into the study, only normal or MCI diagnoses were made at baseline. MCI was diagnosed using international consensus criteria [2]: (a) subjective complaint of decline in memory or other cognitive domain which may be self- or informant-reported; (b) objective impaired performance on cognitive testing, determined by performance on at least one test measure 1.5 SDs or more below published normative values; (c) not demented; (d) normal function or minimal impairment on instrumental activities of daily living [31] attributable to cognitive impairment (Bayer ADL score <3). Participants who met the MCI criteria, but who did not have a self- or informant-reported subjective cognitive complaint were excluded from analysis. At the 6-year follow-up, MCI was diagnosed using identical criteria to baseline and dementia was diagnosed according to the criteria outlined in the Diagnostic and Statistical Manual of Mental Disorders, Fourth Edition (DSM-VI) [32]. Participants with no impairments on neuropsychological tests were deemed to have normal cognition. More details on how clinical diagnoses were made can be found in the S2 Appendix.

## Statistical analysis

To determine the effect of each SCC index on the rate of change in global cognition scores from baseline to 6-year follow-up, we performed a series of linear mixed model analyses. Mixed models are advantageous as they are less prone to non-random attrition bias compared to traditional linear regressions that use only cases with non-missing data [33]. For each model, an SCC index, time-in-study, and the SCC index × time-in-study interaction, were entered as fixed effects. The fixed effect of the interaction term gives the effect of the SCC index on the rate of change of the six cognitive domains plus global cognition over time. Random intercept and slope models were employed with an unstructured variance-covariance matrix of the random effects (the G matrix). Next, we performed a series of Cox proportional-hazards models to see if scores on each SCC index were also associated with the risk of progressing to dementia. For the Cox proportional-hazard models, the time at which progression to dementia occurred was estimated to be at the midway point between the assessment when dementia was first diagnosed and the previous assessment. All mixed models and Cox regression analyses included participants' age, sex and years of education as covariates. To examine the extent to which any associations observed between SCCs and cognition could be due to the common influence of affective symptomology and personality traits, we repeated each analysis with the additional inclusion in the models of participants' GDS, GAS, neuroticism, openness and conscientiousness scores–we refer to these models as partially and fully adjusted, respectively.

# Results

## Sample characteristics

After exclusion criteria were applied, 873 participants and 841 informants were included in the study. Participant characteristics for all predictor variables are presented in Table 1. The mean age of informants was 62.90 (SD = 13.92) and 580 (67.8%) were female.

Table 1. Characteristics of participants at baseline (N = 873).

| Demographics | Values | Min–Max |
|---|---|---|
| Age (years) | 78.65 (4.79) | 70–91 |
| No. of Women (%) | 490 (56.1) | |
| Years of Education | 11.62 (3.50) | 3–24 |
| MMSE[a] | 28.56 (1.33) | 24–30 |
| **Measures of Mood** | | |
| GDS | 2.21 (1.99) | 0–14 |
| GAS | 1.11 (1.89) | 0–8 |
| **Measures of Personality** [b] | | |
| Neuroticism | 15.13 (7.04) | 0–39 |
| Openness | 26.89 (6.02) | 10–43 |
| Conscientiousness | 33.86 (6.05) | 13–48 |
| **SCC Indices** | | |
| Participant Memory | 4.30 (3.00) | 0–14 |
| Participant Non-Memory | 1.32 (1.26) | 0–8 |
| Informant Memory | 2.78 (3.25) | 0–14 |
| Informant Non-Memory | 0.44 (0.78) | 0–4 |

All values are means and standard deviations in parentheses unless otherwise noted.

[a] Total, adjusted for age and education

[b] N = 786

**Table 2. Mixed effect models predicting decline in cognitive domain scores as a function of fixed and random effects.**

| Cognitive Domains | Models 1–4 Partially Adjusted[a] | | | Models 5–8 Fully Adjusted[b] | | |
|---|---|---|---|---|---|---|
| **Global Cognition** | β | SE | p | β | SE | p |
| Participant Memory * Time | -.009 | .004 | .023 | -.009 | .004 | .029 |
| Participant Non-Memory * Time | -.015 | .010 | .117 | -.016 | .010 | .122 |
| **Informant Memory * Time** | **-.016** | **.004** | **< .001** | **-.017** | **.004** | **< .001** |
| Informant Non-Memory* Time | -.025 | .016 | .126 | -.025 | .017 | .138 |
| **Attention Processing Speed** | | | | | | |
| Participant Memory * Time | -.006 | .005 | .190 | -.006 | .005 | .223 |
| Participant Non-Memory * Time | -.013 | .011 | .228 | -.014 | .012 | .212 |
| Informant Memory * Time | -.010 | .004 | .032 | -.011 | .005 | .022 |
| Informant Non-Memory* Time | -.015 | .018 | .404 | -.019 | .019 | .318 |
| **Visuospatial** | | | | | | |
| Participant Memory * Time | -.003 | .004 | .417 | -.003 | .004 | .512 |
| Participant Non-Memory * Time | -.004 | .009 | .659 | -.006 | .009 | .511 |
| Informant Memory * Time | -.008 | .004 | .031 | -.007 | .004 | .054 |
| Informant Non-Memory* Time | .014 | .015 | .342 | .016 | .016 | .306 |
| **Language** | | | | | | |
| Participant Memory * Time | **-.011** | **.004** | **.002** | **-.011** | **.004** | **.005** |
| Participant Non-Memory * Time | -.014 | .009 | .124 | -.014 | .009 | .138 |
| Informant Memory * Time | -.006 | .004 | .073 | -.007 | .004 | .056 |
| Informant Non-Memory* Time | -.010 | .015 | .483 | -.013 | .016 | .417 |
| **Executive Function** | | | | | | |
| Participant Memory * Time | -.007 | .005 | .160 | -.007 | .005 | .170 |
| Participant Non-Memory * Time | -.022 | .012 | .061 | -.023 | .012 | .058 |
| Informant Memory * Time | **-.013** | **.005** | **.009** | **-.014** | **.005** | **.006** |
| Informant Non-Memory* Time | -.016 | .020 | .431 | -.019 | .021 | .371 |
| **Memory** | | | | | | |
| Participant Memory * Time | -.006 | .004 | .118 | -.007 | .004 | .078 |
| Participant Non-Memory * Time | -.004 | .009 | .706 | -.007 | .010 | .492 |
| **Informant Memory * Time** | **-.012** | **.004** | **.002** | **-.013** | **.004** | **.001** |
| Informant Non-Memory * Time | -.025 | .016 | .117 | -.030 | .017 | .072 |

β's are unstandardized regression coefficients

[a] Regressions were controlled for participant age, sex and education.

[b] Regressions were controlled for participant age, sex, education GDS, GAS, neuroticism, openness and conscientiousness scales of the NEO-FFI.

Values that are bold indicate significance at $p \leq .01$.

## Do SCCs predict cognitive decline?

Table 2 presents the results of a series of linear mixed models conducted to determine whether each of the 4 SCC indices was predictive of the rate of change in global cognition scores, as well as the rate of change across the five individual cognitive domains: attention processing speed, visuospatial ability, language, executive function, and memory. For all cognitive domains, we tested each individual SCC index in a partially adjusted model, which controlled for age, sex and education, and again in a fully adjusted model, which controlled for the additional covariates of mood and personality; the result being eight individual models for global cognition and each cognitive domain. In each table, partially adjusted models are labelled Models 1–4 and fully adjusted models are labelled Models 5–8.

**Table 3. Cox proportional hazard regression models of incident dementia over 6 years for individual SCC models.**

| Individual SCC Models | Models 1–4 Partially Adjusted[a] | | | Models 5–8 Fully Adjusted[b] | | |
|---|---|---|---|---|---|---|
| | HR | 95% CI | P | HR | 95% CI | P |
| Participant Memory | 1.08 | 1.01–1.17 | .027 | 1.08 | 1.00–1.17 | .057 |
| Participant Non-Memory | 1.13 | 0.95–1.34 | .164 | 1.10 | 0.91–1.35 | .301 |
| Informant Memory | 1.18 | 1.12–1.25 | < .001 | 1.20 | 1.12–1.28 | < .001 |
| Informant Non-Memory | 1.44 | 1.14–1.81 | .002 | 1.45 | 1.13–1.86 | .003 |

HR = hazard ratio

[a] Regressions were controlled for participant age, sex and education.

[b] Regressions were controlled for participant age, sex, education GDS, GAS, neuroticism, openness and conscientiousness scales of the NEO-FFI.

Our primary outcome measure was global cognition. When controlling for age, sex and education only, participant memory SCCs ($p = .023$) and informant memory SCCs ($p < .001$) significantly predicted the rate of decline in global cognition over 6-years. When controlling for the additional covariates of mood and personality, both informant ($p < .001$) and participant memory SCCs ($p = .029$) remained significant, though informant memory SCCs were more strongly associated with the rate of decline in both the partially and fully adjusted models.

Next, we considered SCCs relationship with rate of decline across the five individual cognitive domains. For attention processing speed, informant memory SCCs significantly predicted the rate of decline in both the partially ($p = .032$) and fully ($p = .022$) adjusted models. For visuospatial, informant memory SCCs significantly predicting the rate of decline in the partially adjusted model ($p = .032$) and neared significance in the fully adjusted model, ($p = .054$). For language, participant memory SCCs significantly predicted the rate of decline in both the partially ($p = .002$) and fully ($p = .005$) adjusted models. Informant memory SCCs approached significance in the fully adjusted model only ($p = .056$) for this domain. For executive function, again, informant memory SCCs were associated with rate of decline in both the partially ($p = .009$) and fully ($p = .006$) adjusted models and participant memory SCCs neared significance ($p = .058$). Finally, for memory, informant memory SCCs significantly predicted the rate of decline in both the partially ($p$ .002) and fully ($p = .001$) adjusted models; participant memory SCCs were non-significant in predicting memory decline. After Bonferroni correction for multiple testing was made for the secondary analyses of the five individual cognitive domains (adjusted critical $p = .05/5 = .01$), participant memory SCCs still significantly predicted the rate for language decline in both the partially and fully adjusted models, and informant memory SCCs significantly predicted the rate of decline for executive function and memory in both the partially and fully adjusted models. Neither participant nor informant non-memory SCCs were associated with the rate of decline for global cognition or any of the five individual cognitive domains.

## Do SCCs predict incident dementia?

Six years after baseline, of 82 participants diagnosed with dementia, 44 (53.7%) were diagnosed with AD, 26 (31.7%) with mixed dementia, 7 (8.5%) with vascular dementia, 2 (2.4%) with Parkinson's dementia, and 1 with Lewy body dementia. Table 3 presents the results of a series of Cox partially and fully adjusted proportional hazard regression models predicting risk of progression to dementia. In the partially adjusted models (Models 1–4), participant memory ($p = .027$), informant memory ($p < .001$), and informant non-memory ($p = .002$) SCCs predicted a

greater risk of dementia. In the fully adjusted models (Models 5–8), informant memory ($p <$ .001)) and non-memory ($p$ = .003) SCCs remained significantly associated with a higher risk of conversion to dementia, with participant memory SCCs trending towards significance ($p$ = .057)

## Discussion

This study explored the relationship between participant and informant memory and non-memory SCCs, cognitive decline and incident dementia over six years while controlling for potentially confounding variables such as participant mood and personality. We found that both participant and informant memory-specific SCCs were associated with a steeper rate of decline in global cognition scores over 6 years, even after controlling for potential confounders such as mood and personality. That is, those who had more self- or informant-reported SCCs that were memory specific at baseline declined more quickly in global cognition. This is note-worthy given that several studies [e.g., 34] argued that SCCs simply reflect an individual's current affective status, or a general tendency towards stress and rumination, rather than foreshadow future cognitive decline.

We also considered the relationship between SCCs and the individual cognitive domains that make up global cognition. In general, informant memory-specific SCCs were most strongly associated with a faster rate of decline across all cognitive domains apart from language. After correcting for multiple comparisons, informant memory SCCs still significantly predicted the rate of decline for executive function and memory in both the partially and fully adjusted models. Participant memory-specific SCCs were significantly associated with the rate of decline for language in both partially and fully adjusted models, but in general were not as strongly associated with decline across the other four domains. Interestingly, many clinicians report patients often complain of experiencing "memory" difficulties when they are experiencing word finding difficulties [35]. Thus, this association may be a result of a lack of general insight into what qualifies memory difficulty from a clinical point of view. Finally, neither participant- nor informant-reported non-memory SCCs were associated with decline in global cognition or the five individual cognitive domains. Taken together, these results suggest that both self-reported and informant-reported memory-specific SCCs may be an early warning sign of cognitive decline globally. Further, self-reported memory decline may indicate changes in language and informant-reported memory decline may be more sensitive to executive function and memory changes over time.

Using Cox proportional hazard analyses to account for attrition bias, participant memory, informant memory and informant non-memory SCCs predicted a greater risk of dementia in the partially adjusted models. However, after fully adjusting for mood and personality, only informant memory and non-memory SCCs remained significant. One explanation for this pattern might be that mood and personality account at least partially for the associated risk between participant memory SCC's and incident dementia. That is, individuals at higher risk of dementia in 6 years may be those with lower mood and certain personality traits who also report memory-specific SCCs. However, it is not clear from our data whether low mood and personality traits like neuroticism contribute to the risk of incident dementia or are secondary to an individual noticing their own cognitive decline. Informant complaints on the other hand do have specific predictive ability for cognitive decline and incident dementia over and above participant mood/personality.

Finally, our findings provide partial support for the new Subjective Cognitive Decline *plus* (SCD *plus*) framework [36]. Specifically, the SCD plus framework states that subjective decline in memory, rather than other domains of cognition, increase the likelihood of preclinical AD. Our results partially support this definition as participant non-memory SCCs were not related

to decline in global cognition or any of the individual cognitive domains; however, informant non-memory SCCs did predict incident dementia. Second, we found less support for the utility of subjective complaints by the individual and greater support for informant reports as predictors of decline. This too is in line with the SCC-I's suggestion to include confirmation by an informant as an SCC *plus* feature because it may serve as an enrichment strategy for preclinical AD, particularly at the progressed stage of SCC [36].

Though our study had many strengths, including a large, well-characterized sample, participant- and informant-reported complaint indices, and memory- and non-memory specific SCCs, there were several limitations. First, for simplicity and to keep with our group's past research [i.e., 20], all cognitive complaint scores were dichotomized such that the complaint was considered present or not present, and then summed to form each of the SCC indices. Dichotomizing complaint scores did not allow for the assessment of SCC severity on a continuous scale, even though, more items endorsed could be construed as a measure of severity. A second limitation is that the number of items in each SCC index varied from 15 (participant and informant memory), to 9 (participant non-memory) to 4 (informant non-memory) questions. Because each SCC index score was made up of a sum of the number of questions endorsed, a one-unit increase in one index is not comparable to a one-unit increase in another index. So, regression coefficients or HRs from analyses with different indices do not give a direct comparison of effect strengths.

## Conclusions

The current findings are an important step in understanding the complex relationships between the subjective experiences of cognitive decline and objectively measured cognitive changes. Our results support several other reviews and metanalyses [e.g., 6, 37, 38] that have shown SCCs, although usually weakly associated with current cognitive impairment, are predictive of future cognitive decline, especially with longer follow-up. Indeed, our results showed a stronger association with cognitive decline and incident dementia compared to earlier work from our group, which found no associations cross-sectionally [19] and weak associations after 2 years. [20] Further, though several studies have argued that SCCs are more consistently associated with mood and personality than cognitive impairment, we found participant and informant memory-specific SCCs predicted decline in global cognition even after controlling for these variables. However, in terms of predicating incident dementia, only informant memory SCCs remained significant after controlling for mood and personality; participant memory SCCs were significant in the partially adjusted model. This suggests lower mood and certain personality traits may be accounting at least some of the relationship between participant SCCs and incident dementia.

From a real-world clinical perspective, when older adults present to their general practitioner with memory-specific cognitive complaints, it would be prudent to take this seriously as they are associated with decline in global cognition over 6-years and may be predictive of incident dementia, particularly if the patient is depressed or anxious and/or has particular personality traits. Further, if and where possible, informants should be sought and asked to report on their perceptions of the patient's memory ability and any memory-specific changes that they have noticed as these increase the index of diagnostic suspicion.

## Supporting information

**S1 Appendix. Subjective complaints questions.** Paraphrased SCC question followed by response options in parentheses.
(DOCX)

**S2 Appendix. Explanatory notes for MAS diagnostic classifications.** Diagnostic criteria used for normal, MCI and dementia consensus diagnoses.
(DOCX)

**S3 Appendix. Explanatory notes for MAS cognitive domain and global cognition scores.**
(DOCX)

**S1 Table. Neuropsychological tests used to calculate cognitive domains and global cognition scores.**
(DOCX)

# Acknowledgments

We thank the participants for their enthusiastic support. The Sydney Memory and Ageing Study Team comprises, in addition to the authors, the following individuals (research assistants, past study coordinators, data assistants): Adam Bentvelzen, Virginia Winter, Josie Bigland, Allison Bowman, Kim Burns, Anthony Broe, Joula Dekker, Louise Dooley, Michele de Permentier, Sarah Fairjones, Janelle Fletcher, Therese French, Cathy Foster, Emma Nugent-Cleary-Fox, Chien Gooi, Evelyn Harvey, Rebecca Helyer, Sharpley Hsieh, Laura Hughes, Sarah Jacek, Mary Johnston, Kate Maston, Donna McCade, Samantha Meeth, Eveline Milne, Angharad Moir, Ros O'Grady, Kia Pfaeffli, Carine Pose, Simone Reppermund, Laura Reuser, Amanda Rose, Peter Schofield, Zeeshan Shahnawaz, Amanda Sharpley, Melissa Slavin, Adam Theobald, Claire Thompson, Ruby Tsang, Wiebke Queisser, and Sam Wong.

# Author Contributions

**Conceptualization:** Katya Numbers.

**Data curation:** Katya Numbers.

**Formal analysis:** Katya Numbers.

**Supervision:** Henry Brodaty.

**Writing – original draft:** Katya Numbers.

**Writing – review & editing:** John D. Crawford, Nicole A. Kochan, Brian Draper, Perminder S. Sachdev, Henry Brodaty.

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
