## [Decision Letter · Decision Letter 0]

23 Mar 2020

PONE-D-20-00404

Participant and Informant Memory-Specific Cognitive Complaints Predict Future Decline and Incident Dementia: Findings from the Sydney Memory and Ageing Study

PLOS ONE

Dear Katya,

Thank you for submitting your manuscript to PLOS ONE. After careful consideration, we feel that it has merit but does not fully meet PLOS ONE’s publication criteria as it currently stands. Therefore, we invite you to submit a revised version of the manuscript that addresses the points raised during the review process. Please address reviewer 1 concerns and resubmit. 

We would appreciate receiving your revised manuscript by May 1, 2020. To enhance the reproducibility of your results, we recommend that if applicable you deposit your laboratory protocols in protocols.io, where a protocol can be assigned its own identifier (DOI) such that it can be cited independently in the future. For instructions see: http://journals.plos.org/plosone/s/submission-guidelines#loc-laboratory-protocols

We look forward to receiving your revised manuscript.

Kind regards,

Hemachandra Reddy

Academic Editor

PLOS ONE

Journal Requirements:

Reviewers' comments:

Reviewer's Responses to Questions

**Comments to the Author**

1. Is the manuscript technically sound, and do the data support the conclusions?

Reviewer #1: Yes

Reviewer #2: Yes

2. Has the statistical analysis been performed appropriately and rigorously? 

Reviewer #1: Yes

Reviewer #2: Yes

3. Have the authors made all data underlying the findings in their manuscript fully available?

Reviewer #1: Yes

Reviewer #2: Yes

4. Is the manuscript presented in an intelligible fashion and written in standard English?

Reviewer #1: Yes

Reviewer #2: Yes

5. Review Comments to the Author

Reviewer #1: The study presents the results of original research. The findings of the manuscript are already well-known but data presented here help put what is known into a little more focus (patients perceive language difficulties, family perceives executive problems). Experiments, statistics, and other analyses are performed to a high technical standard and are described in sufficient detail. Conclusions are presented in an appropriate fashion. The article is presented in an intelligible fashion and is written in standard English. The research meets all applicable standards for the ethics of experimentation and research integrity. The article adheres to appropriate reporting guidelines and community standards for data availability.

Just one thing worth addressing that does not quite make sense and which is important to the logic of this study: if informants are corroborating the presence of deficits, then aren't the deficits by definition objective rather than subjective?

Reviewer #2: I believe this is the first study to examine the relationship between participant-reported and informant-reported memory and non-memory SCCs, cognitive decline and incident dementia over six years. The methods are well executed and well written manuscript.

6. PLOS authors have the option to publish the peer review history of their article (what does this mean?). If published, this will include your full peer review and any attached files.

Reviewer #1: No

Reviewer #2: No

---

## [Author Response · Author response to Decision Letter 0]

20 Apr 2020

Reviewer #1: The study presents the results of original research. The findings of the manuscript are already well-known, but data presented here help put what is known into a little more focus (patients perceive language difficulties, family perceives executive problems). Experiments, statistics, and other analyses are performed to a high technical standard and are described in sufficient detail. Conclusions are presented in an appropriate fashion. The article is presented in an intelligible fashion and is written in standard English. The research meets all applicable standards for the ethics of experimentation and research integrity. The article adheres to appropriate reporting guidelines and community standards for data availability. 

Just one thing worth addressing that does not quite make sense and which is important to the logic of this study: if informants are corroborating the presence of deficits, then aren't the deficits by definition objective rather than subjective?

Reviewer #2: I believe this is the first study to examine the relationship between participant-reported and informant-reported memory and non-memory SCCs, cognitive decline and incident dementia over six years. The methods are well executed and well written manuscript.

Regarding use of the term “subjective” when others have also reported cognitive decline, we point out that we use this term in line with MCI consensus criteria (Winblad et al., 2004), DSM-5 definitions (American Psychological Association, 2013) and the more recently published SCD consensus criteria paper (Jessen et al., 2014). “Objective impairment”, by convention, is determined by performance on validated psychometric tools using established cut-off scores adjusted for age and education, where “subjective impairment” is any self-reported or other-reported impression of impairment or decline. 

The comment by Reviewer 1 also suggests that the participants’ complaints were corroborated by the responses of their informants, however, we examined informant and participant complaints and their respective predictive values separately in this paper. That is, no analyses were carried out to determine whether there was a high level of agreement between informants’ and participants’ subjective evaluations of cognitive ability. Although we agree that examining consensus between informant and participant evaluations is an interesting research question, and worthy of investigation in a future paper based on this cohort, in the present paper no analyses were carried out to examine whether this was, in fact, the case. For these reasons, we have not amended the manuscript.

---

## [Decision Letter · Decision Letter 1]

27 Apr 2020

Participant and informant memory-specific cognitive complaints predict future decline and incident dementia: Findings from the Sydney Memory and Ageing Study.

PONE-D-20-00404R1

Dear Dr. Katya Terra,

We are pleased to inform you that your manuscript has been judged scientifically suitable for publication and will be formally accepted for publication once it complies with all outstanding technical requirements.

With kind regards,

Hemachandra Reddy

Academic Editor

PLOS ONE

Additional Editor Comments (optional):

Reviewers' comments:

Reviewer's Responses to Questions

**Comments to the Author**

1. If the authors have adequately addressed your comments raised in a previous round of review and you feel that this manuscript is now acceptable for publication, you may indicate that here to bypass the “Comments to the Author” section, enter your conflict of interest statement in the “Confidential to Editor” section, and submit your "Accept" recommendation.

Reviewer #1: All comments have been addressed

Reviewer #2: All comments have been addressed

2. Is the manuscript technically sound, and do the data support the conclusions?

Reviewer #1: Yes

Reviewer #2: Yes

3. Has the statistical analysis been performed appropriately and rigorously? 

Reviewer #1: Yes

Reviewer #2: Yes

4. Have the authors made all data underlying the findings in their manuscript fully available?

Reviewer #1: Yes

Reviewer #2: Yes

5. Is the manuscript presented in an intelligible fashion and written in standard English?

Reviewer #1: Yes

Reviewer #2: Yes

6. Review Comments to the Author

Reviewer #1: no additional comments no additional comments no additional comments no additional comments no additional comments

Reviewer #2: (No Response)

7. PLOS authors have the option to publish the peer review history of their article (what does this mean?). If published, this will include your full peer review and any attached files.

Reviewer #1: No

Reviewer #2: No

---

## [Editor Report · Acceptance letter]

30 Apr 2020

PONE-D-20-00404R1 

Participant and informant memory-specific cognitive complaints predict future decline and incident dementia: Findings from the Sydney Memory and Ageing Study. 

Dear Dr. Numbers:

I am pleased to inform you that your manuscript has been deemed suitable for publication in PLOS ONE. Congratulations! Your manuscript is now with our production department. 

With kind regards,

on behalf of

Dr. Hemachandra Reddy 

Academic Editor

PLOS ONE